# Polyphenol Profiling of Chestnut Pericarp, Integument and Curing Water Extracts to Qualify These Food By-Products as a Source of Antioxidants

**DOI:** 10.3390/molecules26082335

**Published:** 2021-04-17

**Authors:** Gabriella Pinto, Sabrina De Pascale, Maria Aponte, Andrea Scaloni, Francesco Addeo, Simonetta Caira

**Affiliations:** 1Department of Chemical Sciences, University of Naples “Federico II”, via Cintia, 80126 Naples, Italy; gabriella.pinto@unina.it; 2Proteomics & Mass Spectrometry Laboratory, ISPAAM, National Research Council, via Argine 1085, 80147 Naples, Italy; sabrina.depascale@ispaam.cnr.it (S.D.P.); andrea.scaloni@cnr.it (A.S.); 3Dipartimento di Agraria, Università degli Studi di Napoli “Federico II”, via Università 100, Parco Gussone, 80055 Portici, Italy; maria.aponte@unina.it (M.A.); addeo@unina.it (F.A.)

**Keywords:** chestnut, water curing, non-targeted MS analysis, antioxidants

## Abstract

Plant polyphenols have beneficial antioxidant effects on human health; practices aimed at preserving their content in foods and/or reusing food by-products are encouraged. The impact of the traditional practice of the water curing procedure of chestnuts, which prevents insect/mould damage during storage, was studied to assess the release of polyphenols from the fruit. Metabolites extracted from pericarp and integument tissues or released in the medium from the water curing process were analyzed by matrix-assisted laser desorption ionization-time of flight-mass spectrometry (MALDI-TOF-MS) and electrospray-quadrupole-time of flight-mass spectrometry (ESI-qTOF-MS). This identified: (i) condensed and hydrolyzable tannins made of (epi)catechin (procyanidins) and acid ellagic units in pericarp tissues; (ii) polyphenols made of gallocatechin and catechin units condensed with gallate (prodelphinidins) in integument counterparts; (iii) metabolites resembling those reported above in the wastewater from the chestnut curing process. Comparative experiments were also performed on aqueous media recovered from fruits treated with processes involving: (i) tap water; (ii) tap water containing an antifungal *Lb. pentosus* strain; (iii) wastewater from a previous curing treatment. These analyses indicated that the former treatment determines a 6–7-fold higher release of polyphenols in the curing water with respect to the other ones. This event has a negative impact on the luster of treated fruits but qualifies the corresponding wastes as a source of antioxidants. Such a phenomenon does not occur in wastewater from the other curing processes, where the release of polyphenols was reduced, thus preserving the chestnut’s appearance. Polyphenol profiling measurements demonstrated that bacterial presence in water hampered the release of pericarp metabolites. This study provides a rationale to traditional processing practices on fruit appearance and qualifies the corresponding wastes as a source of bioactive compounds for other nutraceutical applications.

## 1. Introduction

According to FAO statistics (http://www.fao.org/faostat/en/#search/chestnut, accessed on 15 January 2021), Europe is among the three top producers of chestnuts in the world (after Asia and China), with roughly 155 ktons of fruits in 2018, among which more than 30% obtained in Italy. About 65% of the Italian chestnut/marron production is localized in the Campania region (INEA, database on foreign trade), where the typical cultivar, Castagna di Montella, is bred. The latter has been certified as a European Protected Geographical Indication (PGI) product and is exported in the form of fresh whole or peeled fruit, or in its dried forms.

Chestnut is considered a functional food with a rich source of polyphenolic compounds, phenolic acids (such as gallic acid) and tannins (primarily ellagic acid) [1,2]. Recently, a comprehensive review has dealt with the elucidation of the high biological value also of the chestnut pericarp and integument (outer and inner shell), which are discarded as by-products during food manufacturing, as reported by the extensive literature on the corresponding in vitro and in vivo bioactive properties [3]. For example, chestnut shell hydroalcoholic extracts tested on several human cell lines were suggested to display potential anti-angiogenic and anti-inflammatory effects [4]. When assayed on trout blood and intestinal leukocytes, they also elicited a stimulatory response of the immune system against possible infectious agents [5]. Several studies on rat models of diabetes fed with chestnut extracts demonstrated a diet-dependent increase of pancreatic cell viability [6] and a corresponding cytoprotective response against hepatorenal injury [7].

Actually, the bioavailability of polyphenols is an important topic of debate [8], which must consider the possible interaction of these metabolites with other food macromolecules as well as the impact of gastrointestinal digestion of these compounds and their subsequent enterocyte absorption. Indeed, the gastrointestinal digestion of polyphenols seems to be the main limiting physiological factor responsible for the conversion of the glucoside into the aglycone species, especially in the switching from an acid pH value to a basic one, remarkably influencing the corresponding bioavailability process. For example, up to 20% of the initial glucoside polyphenol species were shown to resist the simulated gastrointestinal digestion of coffee grounds [8]. Indeed, only 5–10% of aglycone species was estimated to be absorbed by passive diffusion through the enterocyte membrane of the small intestine when the corresponding glucoside derivatives were digested [9]. On the other hand, the chemical structure of phenolic compounds was demonstrated to be modified from microbiota of the large intestine, thus affecting their bioavailability [10]. These findings suggest that the complexity of physiological events and mechanisms affecting the dietary assumption of polyphenols has not been fully clarified and hasnot yet provided a complete picture of the corresponding molecular bioavailability. Nevertheless, chestnut extracts are nowadays used as natural ingredients for the preparation of various functional foods [1,11,12].

Chestnut pericarp and integument are a rich source of tannins [12,13,14], which have been subclassified into condensed tannins (CTs) and hydrolysable tannins (HTs). The first compounds are derived from the condensation and polymerization of monomeric flavonoid units [15], while the second ones are polyesters containing a sugar moiety linked to gallic and ellagic acids. HTs, in turn, comprise C-glucosidic ellagitannins, which are further subdivided into vescalagin- and castalagin-types depending on the configuration of the OH group at C-1 position of the glycosidic chain [16], and flavono-ellagitannins that are formed after linkage of a flavonoid molecule to ellagitannins, yielding acutissimin A and B. The latter compounds have been isolated from chestnut wastes [13] and have gained significant interest due to their antitumor activity [17].

A main issue in the chestnut industry is the high perishability of harvested fruits during storage, which is associated with fungal/insect contamination and mould development. This is generally faced with thermo-hydrotherapy or cold-water curing treatment. The latter is a traditional, simple, and inexpensive processing practice commonly used in the Campania region, which is based on soaking chestnuts in cold water at a 1:1 to 1:2 ponderal ratio for 3–9 days [18]. After soaking completion, chestnuts are rinsed with running fresh water, and resulting wastewaters are treated in dedicated waste plants. After chestnut curing in cold water, fruit skin loose its natural luster [19]. The efficiency of the cold water curing treatment depends on associated lactic and alcoholic fermentation processes, which reduce the pH value of the medium [18]. In this context, the increase of CO_2_ and CH_3_CHO levels in the fermentation medium was reported to play an active role in preserving chestnuts during subsequent storage. Whether the simple treatment of chestnuts with cold water was described to mostly reduce the development of insect larvae of *Curculio elephas* and, depending on concomitant lactic and alcoholic fermentation processes, having a significant effect on fungal contamination during storage [20], the preventive addition of *Lb. pentosus* strain to cold water used for treatment was suggested generating antimicrobial compounds that inhibit fungal growth/germination [20], thus extending shelf-life and preserving the appearance of fruits.

Based on the above, the chestnut by-products that are obtained from the corresponding industry can be different; they may have potential use as a source of polyphenols for food, nutraceutical, leather, and cosmeceutical applications. At first, chestnut pericarp and integument tissues, which are removed during food processing for obtaining fresh and dried products and/or corresponding flour. They represented about 20% of the total weight of the original fruits and were already used for recycling purposes due to their high content of tannins [12,13,21]. However, a characterization of corresponding polyphenols was obtained only for low-mass components. Secondly, the wastewater from the above-mentioned water curing treatments of chestnuts, which should eventually contain polyphenols released from the above-mentioned fruit tissues whose nature, however, has not been characterized yet. The stasis of these dissolved molecules in wastewater should render it susceptible to air oxidation. Thus, knowledge of their molecular nature seems a prerequisite for the possible consideration of reusing these wastewaters as a source of bioactive molecules.

With the aim of considering chestnut by-products for possible recycling purposes, polyphenols from pericarp and integument tissues as well as from wastewater of different fruit water curing treatments were extracted and subjected to qualitative and quantitative experiments based on dedicated assays and detailed MALDI-TOF-MS and ESI-qTOF-MS characterization. This allowed defining different molecular signatures for the analyzed materials, which were peculiar for them and were also prodromal to the rationalization of the mechanisms ongoing during different water curing treatments.

## 2. Results

### 2.1. Polyphenol Extraction

The eventual release of polyphenols in wastewater from water curing treatments of chestnuts should hypothetically involve molecules present in the peripheric tissue layers of fruits, namely pericarp and integument, which are imbibed with corresponding aqueous media. In order to evaluate the nature of these phenolic compounds in the above-mentioned fruit districts, sampled chestnuts were peeled to recover pericarp and integument tissues, which were then powdered and extracted in parallel with different solvents. Molecular recovery was evaluated considering overall signal-to-noise ratios measured through dedicated MS procedures (see below). The mixture acetonitrile/methanol/water 2:2:1 *v*/*v*/*v* was identified as the optimal one for extraction of polyphenols. Thus, pericarp and integument tissues were extracted under continuous agitation, for 48 h, and recovered material was analyzed with MALDI-TOF-MS and ESI-qTOF–MS procedures.

### 2.2. MALDI-TOF-MS Profiling of Polyphenols from Chestnut Pericarp and Integument Tissues

In order to provide the maximal representation of molecular species occurring in chestnut epicarp tissues, MALDI-TOF-MS analysis of the corresponding extracts was performed in both linear positive and negative ion mode. This analysis gave mass spectra almost superimposable between different samples, which can be summarized with the polyphenol profiles shown in Figure 1.

Distribution of signals within the MALDI-TOF mass spectrum acquired in positive ion mode (Figure 1A) fitted a Gaussian profile, which enabled to guess the presence of oligomeric CTs differing from each other by units of catechins and esterified catechin with gallic acid residue (Δm = 152 mass units) [22]. Signals recorded by MALDI-TOF-MS analysis were tentatively attributed to CTs, like procyanidins. Increments of about 288, and 441 mass units were tentatively attributed to the presence of (epi)catechin, and (epi)catechin gallate units, respectively, representing the building block of a wide variety of tannins, from simple monomers to multiple oligomers [23]. The precursor compound corresponded to the epicatechin gallate (*m*/*z* 442.9), with dominant signals at *m*/*z* 426.9 and 410.9, which were suggestive of the loss of one or two OH groups from the B-ring, respectively. This finding was consistent with the presence of fisetinidin, already identified in chestnut [12], which does not present a -OH group at position C3 of one of the two B-rings.

MALDI-TOF-MS analysis of the same sample carried out in negative ion mode showed a quite different molecular profile (Figure 1B). The dominant mass signal at *m*/*z* 933.1 indicated the presence of a chestnut-specific HT, namely castalagin, together with its molecular dimer (at *m*/*z* 1865.2). The occurrence of castalagin in the chestnut pericarp was also confirmed by the co-presence in the mass spectrum of the aglycon castalin (*m*/*z* 631.1), which is one of the main phenolic compounds present in chestnut shells [16]. The loss of two condensed gallic acid units linked via a galloyl ester bond (Δm = about 152 mass units) between two consecutive product ions likely arose from a different isobaric precursor ion at *m*/*z* 631.1 (Figure 1B). Other signals present at *m*/*z* 783.1, 1083.1, and 1101.1 were tentatively assigned to pedunculagin II (bis-HHDP-hex), punicalagin (HHDP-gallagyl-hex), and punicalagin-like species, as already described in other fruits [24,25].

In order to provide the maximal representation of molecular species occurring in chestnut integument tissues, MALDI-TOF-MS analysis of the corresponding extracts was also performed in both linear positive and negative ion mode (Figure 2). Similar to their pericarp counterparts, the mass profiling results for integument extracts showed the occurrence of phenolic polymers.

In the case of the MALDI-TOF mass spectrum acquired in positive linear ion mode, corresponding signals differed for the occurrence of repeated units of epigallocatechin and epicatechin gallate residues (Δm = about 304 and 441 mass units) (Figure 2A). Apart from the MH^+^ signal of epicatechin gallate (*m*/*z* 442.8), the most abundant peak was tentatively assigned to prodelphinidin B3 (*m*/*z* 610.8), a gallocatechin oligomer. MALDI-TOF-MS analysis of the same sample carried out in negative ion mode highly differed from that recorded under the same experimental conditions for pericarp tissues and confirmed results obtained in positive ion mode (Figure 2B). It reflected the occurrence of prodelphinidin oligomers up to nonamers, together with more complex polymers ascribed to HTs, as already found in chestnut bark [26]. In conclusion, dominant MALDI-TOF-MS signals in integument tissues were associated with oligomeric prodelphinidins, which may have hidden peaks of ellagitannins detected in the chestnut shell as a result of signal ion suppression phenomena.

### 2.3. ESI-qTOF-MS Analysis of Polyphenols from Chestnut Pericarp and Integument Tissues

High-resolution mass spectrometry and tandem mass spectrometry (MS/MS) has already been used for establishing with high confidence the identity of the polyphenolic species present in various fruit extracts [27]. With the aim of confirming the tentative molecular assignment of polyphenolics obtained by MALDI-TOF-MS, the above-mentioned chestnut pericarp and integument extracts were also analyzed by ESI-qTOF-MS and ESI-qTOF-MS/MS (Figure 3). Corresponding ESI-qTOF-MS profiles acquired in positive ion mode are shown in Figure 3A,B, respectively. They provided another view based on a soft ionization technique of the polyphenolic compounds present in pericarp and integument tissues, which was not dependent on the matrix-assisted laser desorption ionization process. Moreover, the possibility of acquiring fragmentation spectra of selected ions allowed associating the recorded polyphenolic profiles with the structure of single compounds. Tandem mass spectra recorded for the most abundant precursor ions present in Figure 3 are shown in Appendix A.

When comparing ESI-qTOF-MS profiles of pericarp and integument extracts in the *m*/*z* range of 400–2500, evident differences were observed between the analyzed tissues, either regarding the nature of detected polyphenols and the number of linked monomeric units (Figure 3). In particular, the integument extract showed more intense signals for polyphenols containing 3 to 5 monomeric units, while dimeric-trimeric compounds seemed more represented in the pericarp counterpart. This result confirmed the signal distribution observed in the MALDI-TOF mass spectra of integument and pericarp extracts when the acquisition was performed in positive ion mode (Figure 1A and Figure 2A). In addition, a combination of the accurate measurement of detected MH^+^ signal values in ESI-qTOF mass spectra, its comparison with polyphenolic structures already detected in chestnut tissues [28] and high-quality fragmentation spectra of most abundant ions present in recorded polyphenolic profiles (Appendix A) allowed for the association of a molecular structure to the detected molecular species (Figure 3). The results of this study are summarized in Figure 4, which provides a list of the polyphenols identified in chestnut pericarp and integument extracts by ESI-qTOF-MS, together with their experimental mass value and, in some cases, corresponding diagnostic fragment ions.

In particular, polyphenol polymerization in pericarp tissues was associated with the progressive incorporation of (epi)catechin units, which determined the formation of oligomeric species differing for a Δm = +288 mass units (Figure 3A and Figure 4). In this case, the most significant oligomeric species were (epi)catechin gallate monomer (*m*/*z* 443.1) and (epi)catechin dimer (*m*/*z* 579.1) (Appendix A), which undergo (epi)catechin addition of up to five molecular units. The assignment of signals at *m*/*z* 731.1, 867.2, 1019.2, 1155.3, 1307.3, 1443.3, 1595.3 and 1747.3 reported in Figure 3A was confirmed by the corresponding fragmentation spectra (Figure 4 and Appendix A). Thus, signals at *m*/*z* 443.1, 731.1, 1019.2, 1307.3, 1595.3 and 1883.3 (Figure 3A) were attributed to type-B procyanidin monogalloylated species classified in the group of CTs (Figure 4) Interestingly, hop extracts have been reported containing similar procyanidin profiles [29]. The presence of these type-B procyanidin oligomers was paralleled by the A-type counterparts, as revealed by the corresponding signals at *m*/*z* 577.1, 865.2 and 1153.2 (Figure 3A and Figure 4). This finding was peculiar of the pericarp extract, which thus showed catechin oligomers having a double type-A inter flavanol linkage at C2→O→C7′ (Figure 4). On the other hand, the low-intensity signals at *m*/*z* 1170.1, 1459.3 and 1747.3 (Figure 3A) were attributed to oligomers of the type-B digalloylated procyanidin series (Figure 4), as also proved by the fragmentation spectrum of the latter species (Appendix A). Finally, the mass signals observed at *m*/*z* 935.1 was definitively assigned to castalagin/vescalagin (Figure 3A and Figure 4), as proved by the corresponding MS/MS spectrum that showed fragments at *m*/*z* 633 and 303 corresponding to vescalin/castalin and ellagic acid, respectively (Appendix A).

Other authors were able to discriminate vescalagin from its isomer, castalagin, by recording fragmented ions in negative ion mode related to galloyl-gallagyl-hexoside and gallagyl species [24,30]. When recorded in positive ion mode, some of the above-cited fragment ions were not detected in the case of the chestnut pericarp extract, while the identification of fragment signals at *m*/*z* 277 and 303 definitively proved the assignment of this molecule to castalagin (Appendix A), as already demonstrated by other authors [31].

Conversely, the ESI-qTOF mass spectrum of integument extract showed the occurrence of molecular ions at *m*/*z* 595.1 and 611.1 (Figure 3B), which were associated with epicatechin-(epi)gallocatechin and gallocatechin-(epi)gallocatechin dimer based on fragmentation spectra, respectively (Appendix A). Their mass difference (Δm = +16 mass units) corresponded to an (epi)catechin to gallocatechin substitution. Starting from these two precursor ions, two series of mass signals occurred in the mass spectrum, which differed from each other in the number of polymerized (epi)gallocatechin units (Δm = +304 mass units), reaching a maximum value of six added molecules for (epi)gallocatechin (*m*/*z* 2115.9) and gallocatechin parents (*m*/*z* 2130.8) (Figure 3B and Figure 4). Assignment of signals at *m*/*z* 898.9, 1202.9, 1506.9 and 1811.2 was confirmed by the corresponding fragmentation spectra (Appendix A). The above-mentioned oligomerization process was also observed starting from (epi)catechin gallate monomer (*m*/*z* 443.1) and (epi)catechin dimer (*m*/*z* 579.1), and involved the polymerization of up to six (*m*/*z* 2267.8) and five (*m*/*z* 2098.9) (epi)gallocatechin units, respectively (Figure 3B and Figure 4). In this case, the assignment of parent (epi)catechin dimer (*m*/*z* 579.1) was confirmed by the corresponding fragmentation spectrum (Appendix A).

In conclusion, ESI-qTOF-MS and ESI-qTOF-MS/MS experiments demonstrated that the pericarp extracts contain CTs composed of (epi)catechin units, also called procyanidins, and HTs such as ellagitannins, while the integument counterparts were rich in condensed tannins characterized by gallocatechin and catechins units condensed with gallate, also known as prodelphinidins.

### 2.4. Time-Course Analysis of Total Phenols Released in Wastewater from Different Chestnut Water Curing Treatments

With the aim of quantitatively evaluating the extraction of the above-mentioned polyphenols as a result of different water curing processes, comparative experiments were also performed on aqueous media recovered from fruit treatment with: (i) tap water containing the above-mentioned antifungal *Lb. pentosus* strain (process A); (ii) tap water (process B); (iii) wastewater recycled from a previous curing treatment (process C). Kinetics of polyphenols migration from soaked fruits into different water curing media resulting from the water curing processes A, B and C was evaluated by comparative TPC measurements on corresponding wastewater samples at 0, 24, 48, 72 and 96 h. Quantitative results are shown in Figure 5. As expected, the fruit release of polyphenols in wastewater samples was time-dependent and progressively increased along with the duration of the water curing process.

The greatest diffusion of polyphenols in wastewater samples occurred for the treatment with tap water (process B) (Figure 5B), for which a polyphenol concentration value of 2.9 mg mL^−1^ was measured at 96 h. In this case, time-course analysis of polyphenols migration into the water curing medium demonstrated that the highest and sudden molecular release occurred at the end of the curing process, namely at 96 h, when more than 90% of the total migration phenomenon was observed.

Conversely, the kinetics of polyphenol migration from soaked fruits into water curing media as result of processes A and C were similar to each other, and highly differed from that of process B (Figure 5); at the end of the treatment, the corresponding molecular recovery values (0.4 and 0.6 mg mL^−1^, respectively) accounted for about one-sixth of the one observed for process B. The appearance, color and luster of recovered chestnuts from processes A and C provided a rationale to data reported in Figure 5, being more intense than those of counterparts from process B (data not shown), in agreement with a preserved maintenance of polyphenols in corresponding fruits. Their appearance resembled that of untreated chestnuts. This was consistent with the widely accepted consideration that the water curing process performed with tap water makes the treated chestnuts lose their natural luster [32].

A comparison of processes A and B, which were both performed in tap water, suggested that the presence of *Lb. pentosus* (in process A), yielding lactic acid in the curing medium and thus lowering the corresponding pH value (Appendix A), was the reason for the corresponding reduced release of polyphenols from the fruit, with respect to the treatment not involving bacterial addition (process B).

Although it was reasonable to speculate that wastewater samples from a previous water curing treatment (process C) at the beginning of treatment (t = 0) should have contained higher levels of polyphenolic compounds than those measurable in counterparts from processes A and B, analysis of corresponding samples showed lower concentration values than those expected (Figure 5). This finding was tentatively associated with a putative molecular precipitation phenomenon occurring during wastewater exposition to air, which involved oxidized polyphenol polymerization and left only a small part of molecules in their native and soluble state. A comparative inspection of wastewater sample centrifugates from processes A, B and C confirmed this hypothesis (data not shown), revealing the formation of evident molecular precipitates only in the latter case. The possible interaction of the above-mentioned polymerizing phenolic compounds with the cell wall of pathogenic microorganisms, eventually determining the formation of metabolite-organism co-precipitates, should represent an interesting process for removing the spoilage microflora from chestnuts [33]. However, this polyphenol-microorganism interaction has to be confirmed by additional experimental evidence, and future studies are needed for this purpose.

### 2.5. Mass Spectrometric Analysis of Polyphenols Released in Wastewater from Different Chestnut Water Curing Treatments

In order to comparatively evaluate the nature of polyphenols present in water curing media resulting from processes A, B and C, corresponding wastewater samples were subjected to molecular profiling analysis through MALDI-TOF-MS and ESI-qTOF-MS, as already described for pericarp and integument extracts. In particular, MALDI-TOF-MS analysis in linear positive (Appendix A) and negative ion mode (Figure 6) of wastewater samples A, B and C showed differences between signal profiles. This finding was particularly evident for spectra acquired in negative ion mode, in which samples from process A showed mass signals that were quite different with respect to that present in counterparts from treatments B and C (Figure 6). A careful inspection of the mass spectra from the latter samples evidenced intense mass signals related to castalagin, castalin, castalagin dimer, punicalagin, punicalagin-like and peduncalagin II species (Figure 6B,C), which were already detected as prominent species in the MALDI-TOF-MS profile acquired in linear negative ion mode of pericarp extracts (Figure 1B). Additional minority signals in the mass spectra of samples B and C (Figure 6B,C) were associated with (epi)catechin-(epi)catechin gallate-(epi)gallocatechin-based polyphenols already detected in the MALDI-TOF-MS profile acquired in linear negative ion mode of integument extracts (Figure 2B). This signal intensity pattern was totally inverted in the case of wastewater samples A, where MH^+^ peaks associated with typical polyphenol species detected in integument extracts were prominent with respect to counterparts associated with metabolites assigned in the pericarp extracts. Negligible spectral differences observed between samples B and C suggested that the nature of extracted, soluble polyphenols between the two treatments was not affected by the pre-existing occurrence of these metabolites in the water curing medium before chestnut treatment. The results reported above suggest that the contribution of pericarp polyphenols to total metabolites present in wastewater samples from water curing processes B and C was more relevant than that ongoing after treatment A. Thus, the occurrence of bacteria in the water-curing medium did not hamper the extraction of integument polyphenols but had a pronounced effect on the solubilization of pericarp polyphenolics. Although with a less evident effect, the above-mentioned differences observed between samples A and samples B-C were confirmed when the same wastewater specimens were analyzed in linear positive ion mode (Appendix A). In this case, spectral variations were mostly associated with signals at *m*/*z* 605.0 and 893.2 in samples from treatment A, which again highlighted the occurrence of different phenolic polymers between analyzed samples.

The above-reported polyphenol mass spectral profile similarities/dissimilarities between wastewater samples were further evidenced when these specimens were analyzed by ESI-qTOF-MS analysis in positive ion mode (Figure 7). Evident signals at about *m*/*z* 579.1 (procyanidin B-type), 617.2, 731.1, 867.1 (procyanidin C1) and 1155.2 (procyanidin tetramer) were observed as molecular signatures of wastewater samples from treatments B and C, which were lacking in counterparts from treatment A. Conversely, peculiar signals at *m*/*z* 416.9, 514.9, 530.9, 666.9, 682.9, 691.5, 702.2, 818.9, 834.9, 859.2, 995.4 and 1011.4 were typical of the ESI-qTOF-MS profile of wastewater from treatment A, which were absent in the counterparts from treatment B and C.

These signals were not assigned to specific metabolites. These results again demonstrated that polyphenol compounds present in the wastewater samples from treatment B and C represented the most abundant polyphenols (procyanidin oligomers) present in pericarp (Figure 3A), and emphasized the significant molecular migration of these compounds from the outer part of the chestnuts into the wastewater following treatments B and C. This was not the case in treatment A, where this migration was hampered, and no signals attributable to known chestnut polyphenols were detected, based on data reported in Figure 4.

## 3. Materials and Methods

### 3.1. Chestnut Sampling and Treatments

Chestnuts from the cultivar Castagna di Montella were subjected to the water curing process based on soaking fruits in water at a 1:2 fruit to water ponderal ratio; treatments were performed at the farm “Malerba Castagne”, which is located in Montella (Avellino), Italy. Three identical batches of fresh chestnuts weighing each about 200 kg were soaked in: (A) tap water with adjunct of *Lb. pentosus* strain OM13 added on the second day of the treatment, which was already identified by the agar well-diffusion assay to exhibit a broad-spectrum antifungal activity [20]; (B) tap water; (C) wastewater from a previous water curing treatment, which was recycled for a new consecutive process. All trials were performed in parallel. Samples of wastewater from treatments A, B and C (100 mL) were randomly collected just after the beginning of the water curing process (t = 0) and after 24, 48, 72 and 96 h, and then subjected to the analytical assays reported below. Corresponding pH values are reported in Appendix A. Original tap water used for different treatments was also stored for subsequent use as control samples.

### 3.2. Quantitative Evaluation of Total Phenolic Compounds in Wastewaters

The Folin−Ciocalteu assay was used for the quantification of total polyphenols content (TPC) in wastewater from treatments A, B and C, as well as in original tap water control samples. To this purpose, a 1 mL-aliquot of wastewater was mixed in stopper test tubes with 1 mL of Folin-Ciocalteu reagent (Sigma-Aldrich Co., St. Louis, MO, USA), followed by the addition of 1 mL of Na_2_CO_3_ (75 g/L) (≥99.9%, Sigma-Aldrich Co., USA). The mixture was kept at room temperature for 90 min, and the corresponding absorbance at 765 nm was measured with a UV-VIS Cary 1-E spectrophotometer (Varian, Palo Alto, USA), with respect to a blank sample. All measurements were compared to a standard curve of 10−100 μg/mL gallic acid (98%, Acros Organics, ThermoFischer Scientific, Waltham, MA, USA), and the results expressed as milligrams of gallic acid equivalents (GAE)/100 g of wastewater extract. All measurements were performed in technical triplicate. Validation of the whole quantification procedure was obtained with two blind samples.

### 3.3. Extraction of Polyphenols from Chestnut Integument, Pericarp and Curing Wastewaters

One kg of randomly collected fresh chestnuts was manually peeled to recover corresponding pericarp and integument tissues, which were powdered separately using a food processor (model 843, Moulinex, Milan, Italy). Depending on their nature, polyphenols have a variable solubility in various polar organic solvents [3]. Thus, different solvents were assayed to obtain in a single step the best extraction of the different classes of polyphenols present in the chestnut shell and integument. The corresponding molecular recovery was evaluated on a quantitative basis using the Folin-Ciocalteu assay (data not shown), but also according to the overall signal-to-noise ratios measured during dedicated MS analysis. In this context, the best procedure combining both parameters for the extraction of polyphenols from the solid parts of chestnuts was identified as the treatment with the mixture acetonitrile/methanol/water (2:2:1, *v*/*v*/*v*), at 20 °C. According to the above-mentioned procedure, 5 g of pericarp and integument tissues were placed in 10 mL-centrifuge tubes with screw cap in triplicate and extracted with 10 mL of the above-mentioned mixture, under continuous agitation, for 48 h. After centrifugation at 5000 rpm, for 20 min, 4 °C, recovered liquid was filtered through a 0.20 μm cellulose acetate filter, transferred to novel tubes and dried under a stream of gaseous N_2_. All samples were then solved in 200 μL of 0.1% TFA, which were applied to a desalting step on a C18 solid-phase column (Seppak Oasis, Waters Corporation, Milford, MA, USA); salts were washed away with aqueous 0.1% TFA as eluent, and phenolic acids were finally eluted with 50% *v*/*v* acetonitrile containing 0.1% TFA. Desalted samples were finally dried under a stream of gaseous N_2_. Chestnut pericarp and integument samples were dissolved in 50% *v*/*v* acetonitrile containing 0.1% TFA (1 μg μL^−1^ final concentration) prior to mass spectrometric analysis.

In parallel, 10 mL-aliquots of the liquid recovered from the chestnut water curing treatments A, B and C were filtered through a 0.20 μm cellulose acetate filter and dried at 30 °C under a stream of gaseous N_2_; all samples were then solved, extracted, and processed according to the method described above, prior to mass spectrometry analysis. Wastewater samples were dissolved in 50% *v*/*v* acetonitrile containing 0.1% TFA (1 μg μL^−1^ final concentration) prior to mass spectrometric analysis.

### 3.4. MALDI-TOF-MS Analysis

Chestnut pericarp and integument, and wastewater samples were analyzed in parallel by MALDI-TOF-MS analysis, using a Voyager DE-Pro spectrometer (PerSeptive BioSystems, Framingham, MA, USA) equipped with an N_2_ laser (k = 337 nm). An identical volume (1 μL) of the samples and the matrix solution, which was prepared by dissolving 10 mg of 2,5-dihydroxybenzoic acid in 1 mL of aqueous 50% *v*/*v* acetonitrile containing 0.1% *v*/*v* TFA, were mixed in parallel, placed onto the instrument target and dried at room temperature. The instrument operated with an accelerating voltage of 20 kV, a grid voltage of 95% of the accelerating voltage, a guidewire of 0.05 % and a delayed-ion extraction time of 175 ns. External mass calibration was performed with Sequazyme™ Peptide Mass Standards Kit, Calibration Mixture 1 and Calibration Mixture 2 (Applied Biosystems, Foster City, CA) containing des-Arg1-bradykinin, angiotensin I, Glu1-fibrinopeptide B, neurotensin 0.2, angiotensin I, ACTH (1–17 clip), ACTH (18–39 clip), ACTH (7–38 clip) and bovine insulin, respectively. The mass spectra were acquired in linear positive and negative ion modes. Raw data were elaborated using the software program Data Explorer version 4.0 (Applied Biosystems, Foster, CA, USA).

### 3.5. ESI-qTOF-MS Analysis

Chestnut pericarp and integument, and wastewater samples were analyzed in parallel by ESI-qTOF-MS analysis, using an ESI q-TOF™ hybrid quadrupole/time-of-flight mass spectrometer (Micromass Ltd., Manchester, UK) equipped with a Z-spray ion source. Dissolved samples were delivered in the instrument source through a syringe pump operating at 0.5 μL min^−1^. Spectral acquisition was performed in positive ion mode and recording MS and MS/MS spectra. The source and desolvation temperature values were 100 °C and 200 °C, respectively. The TOF operated with an acceleration voltage of 9.1 kV, a cone voltage of 100 V, a cone gas (N_2_) of 13 L h^−1^, and the collision energy in MS mode of 10 eV. Collision-induced dissociation (CID) spectra were acquired in a data-dependent method on the most abundant ions having *m*/*z* values ranging from 300 to 2500. The collision energy was dependent on the *m*/*z* value and the charge state of the parent ion; it was generally in the range 25–40 V. The collision cell was pressurized with 10.34 Pa ultrapure Ar (99.999%).3.6. Statistical Analysis

All quantitative measurements reported in the present work were performed in technical triplicate; the calculation of the corresponding mean and standard deviation values were performed with Microsoft Excel 2017 software (Microsoft Corporation, Redmond, WA, USA). All statistical analyses were performed at a significance level of 5% (*p* ≤ 0.05) using XLSTAT software Version 2020.1 (Microsoft Corporation, Redmond, WA, USA).

## 4. Conclusions

This study provides useful indications of the molecular processes associated with the traditional practice of the water curing of chestnuts, which is aimed at preventing insect and mould development during fruit storage. At first, the combined MALDI-TOF-MS and ESI-qTOF-MS analyses characterized different polyphenols present in fruit pericarp and integument tissues. In this context, fragmentation experiments with the latter technique allowed assigning occurring metabolites, demonstrating that CTs and HTs made of repeated catechin/epicatechin and ellagic acid units were highly represented in the fruit pericarp, whereas polymerized gallocatechin or catechin units esterified by gallic acid occurred in the integument tissues. Then, different water curing treatments, namely with tap water, tap water containing *Lb. pentosus* and wastewater from a previous water curing treatment, were compared as it regards to the luster/appearance of recovered chestnuts, as well as the amount and the nature of polyphenols released in the corresponding wastewaters. Results definitively indicated that the wastewater from chestnut water curing treatment can be used as a rich source of polyphenols. It also provided a rationale to the loss of chestnut luster during curing treatment with tap water, demonstrating a strong release of pericarp metabolites in the corresponding wastewaters. This condition was hampered when treatments were performed at conditions involving a drop of the pH value of the medium, namely tap water containing *Lb. pentosus* or wastewater from a previous water curing treatment, which preserved fruit appearance. The nature of the polyphenols detected in wastewaters from treatment with tap water or wastewater from a previous water curing treatment suggested an evident contribution of pericarp polyphenols to metabolites dissolved in wastewaters, which was absent in the case of tap water containing *Lb. pentosus*. In conclusion, this study provides a rationale to different water curing treatments as in terms of the recovered chestnuts but also to optimal conditions for promoting the release of bioactive natural products in wastewaters, thus facilitating their recycling. The latter treatments, favoring the optimal recovery of polyphenols, may find a diffused use when the appearance of treated fruits is not important for commercial purposes, as in the case of raw materials used for production of chestnut flour.

## Figures and Tables

**Figure 1 molecules-26-02335-f001:**
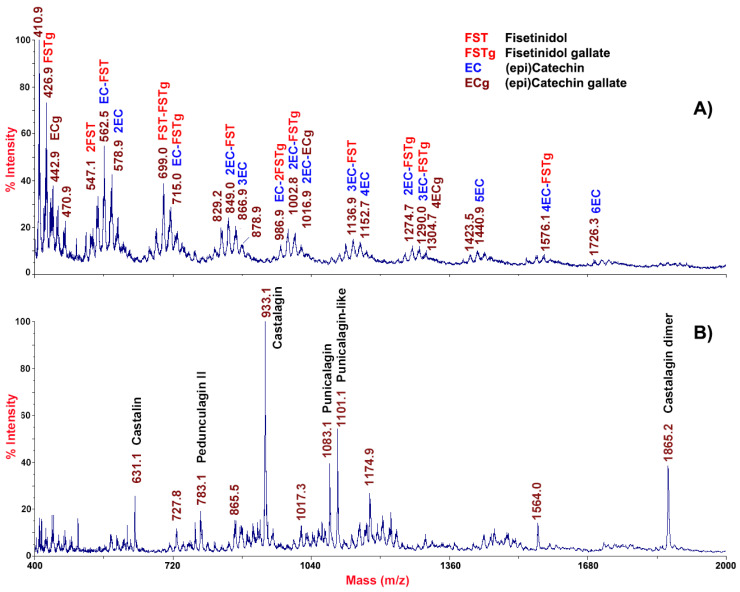
MALDI-TOF mass spectrum of the extract of chestnut pericarp tissues acquired in linear positive (panel (**A**)) and negative (panel (**B**)) ion mode. Condensed polymeric species were tentatively ascribed to gallotannins and ellagitannis derivatives. FST, fisetinidol; FSTg, fisetinidol gallate; EC, (epi)catechin; ECg, (epi)catechin gallate; g, gallic acid.

**Figure 2 molecules-26-02335-f002:**
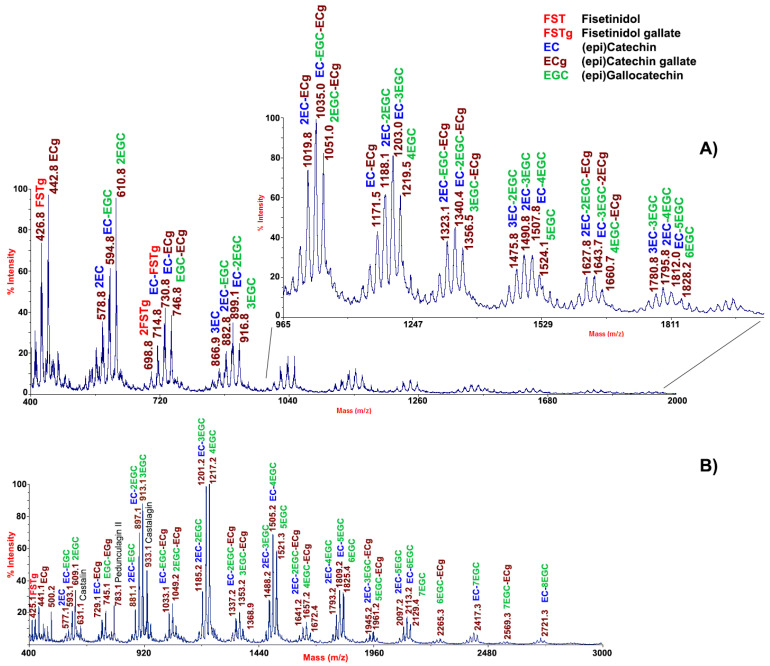
MALDI-TOF mass spectrum of the extract of chestnut integument tissues acquired in linear positive (panel (**A**)) and negative (panel (**B**)) ion mode. FST, fisetinidol; FSTg, fisetinidol gallate; EC, (epi)catechin; ECg, (epi)catechin gallate; EGC, (epi)gallocatechin.

**Figure 3 molecules-26-02335-f003:**
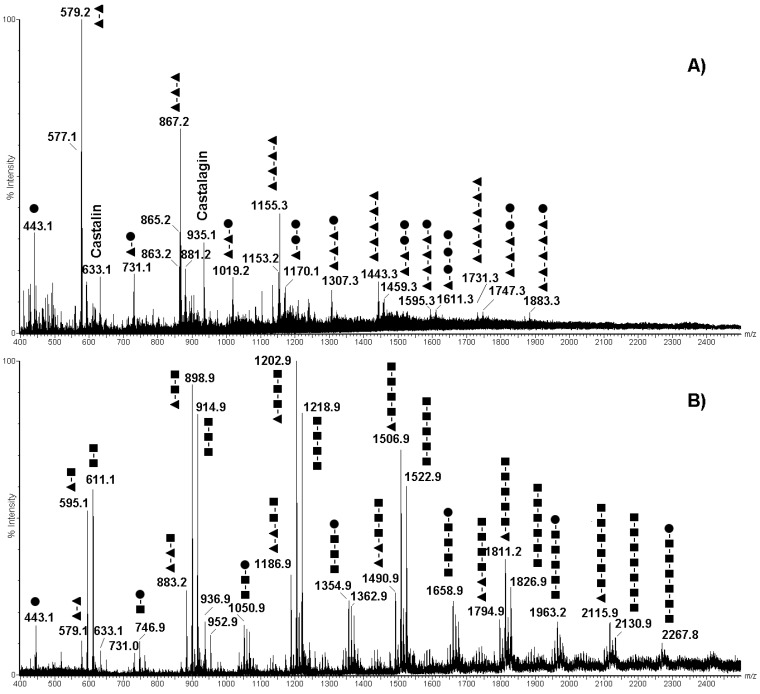
ESI-qTOF mass spectra of extracts from chestnut pericarp (panel (**A**)) and integument (panel (**B**)) tissues acquired in positive ion mode. In panel (**A**), assigned are procyanidins non-galloylated and mono-galloylated oligomers; in panel (**B**), labeled are epigallocatechin and epicatechin gallate oligomers. ▲, epicatechin; ■, epigallocatechin; ●, epicatechin gallate.

**Figure 4 molecules-26-02335-f004:**
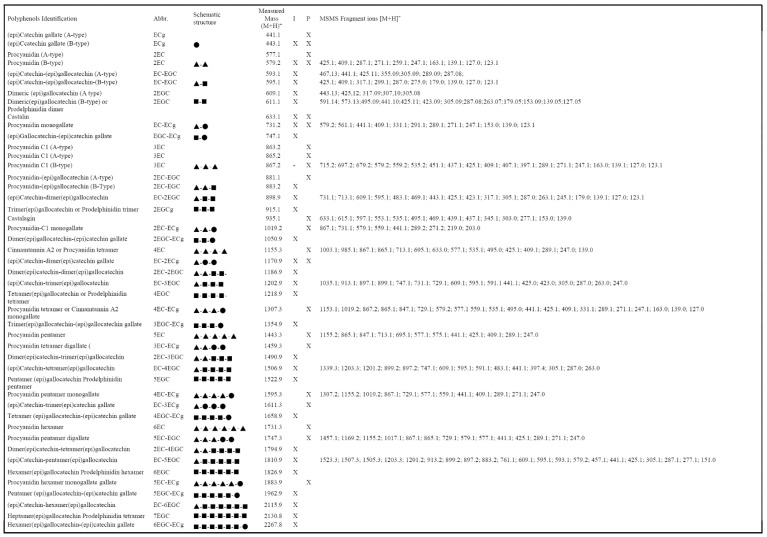
List of polyphenols identified in chestnut pericarp and integument extracts (Figure 3A,B), along with their measured mass values ascertained by ESI-qTOF-MS analysis in positive ion mode and major diagnostic fragment ions recorded for each precursor ions. I, integument; P, pericarp; EC, (epi)catechin, ▲; EGC, (epi)gallocatechin, ■; ECg, (epi)catechingallate, ●.

**Figure 5 molecules-26-02335-f005:**
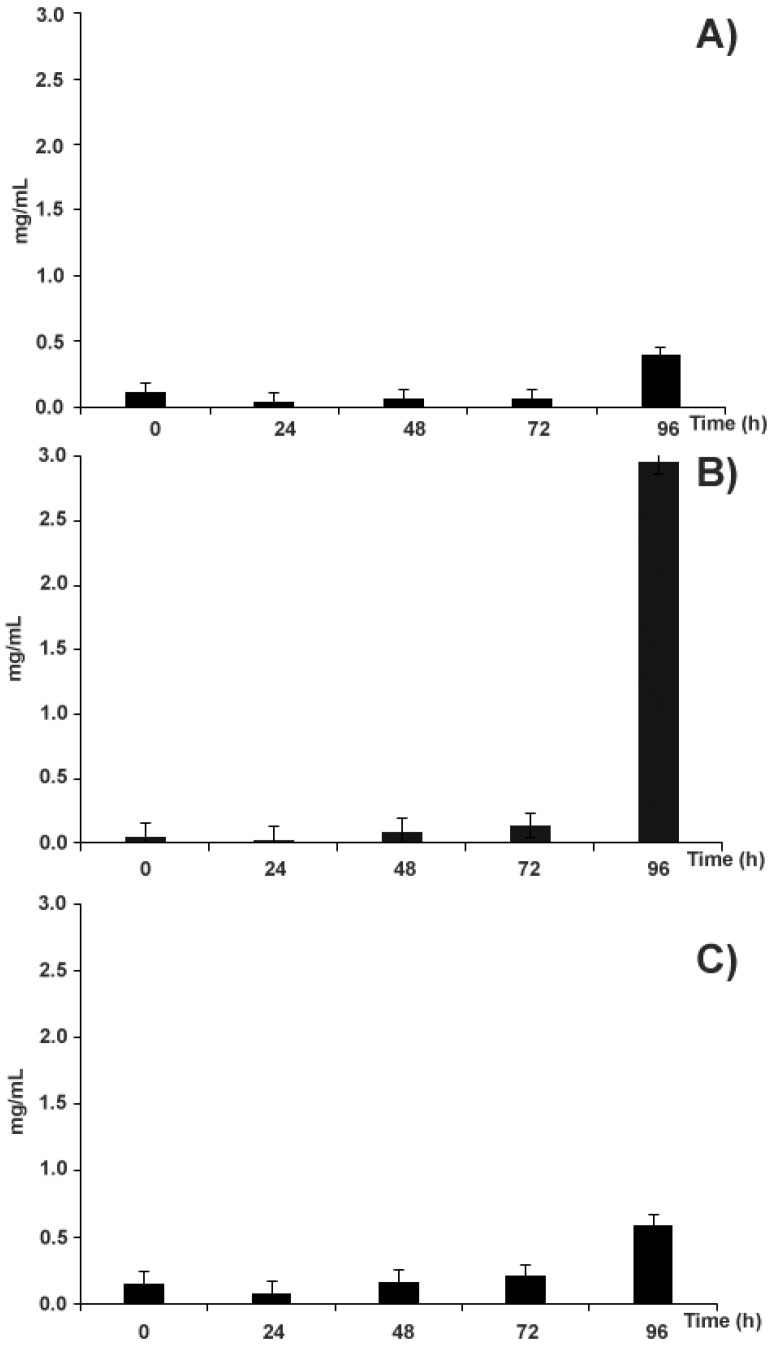
Kinetics of the diffusion of phenolic compounds from the whole fruits into the water-curing media deriving from the chestnut curing processes with: (i) tap water containing cultured *Lb. pentosus* (panel (**A**)); (ii) tap water (panel (**B**)); (iii) wastewater from a previous curing treatment (panel (**C**)). Reported values refer to data from water curing media from different curing processes, which were subtracted from counterparts from blank samples.

**Figure 6 molecules-26-02335-f006:**
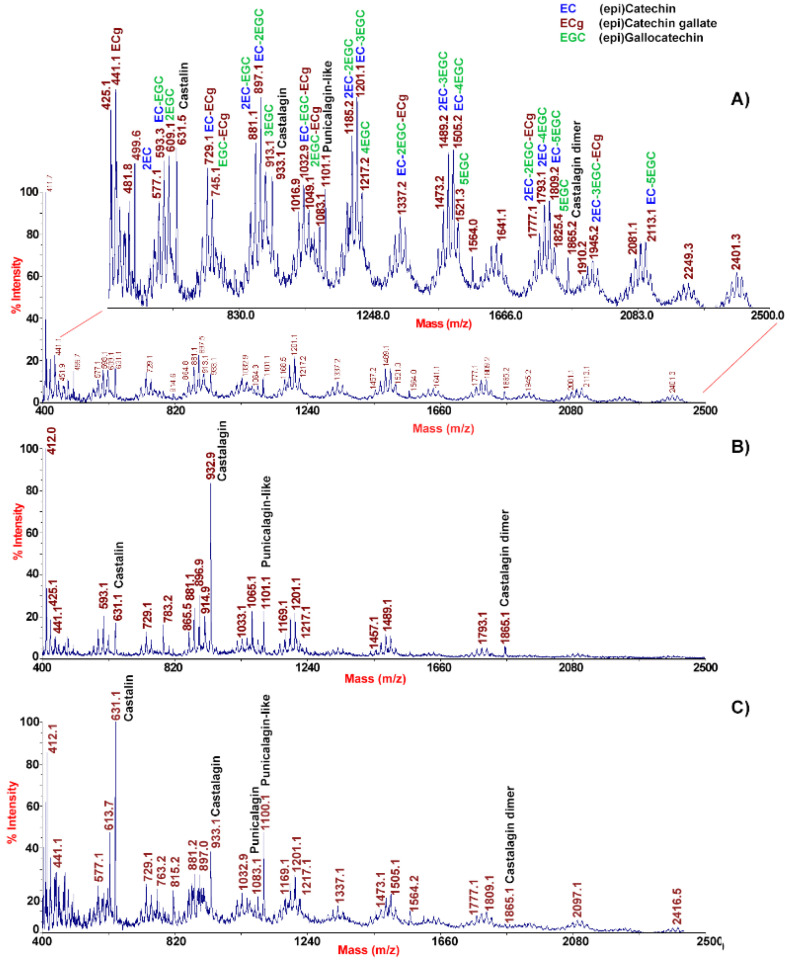
MALDI-TOF mass spectra acquired in linear negative ion mode of wastewater samples resulting from chestnuts subjected to water curing with: (i) tap water containing cultured *Lb. pentosus* (panel (**A**)); (ii) tap water (panel (**B**)); (iii) wastewater from a previous curing treatment with tap water (panel (**C**)). Condensed polymeric species were ascribed to polyphenols based on MALDI-TOF mass spectra acquired in linear negative ion mode of chestnut pericarp and integument extracts.

**Figure 7 molecules-26-02335-f007:**
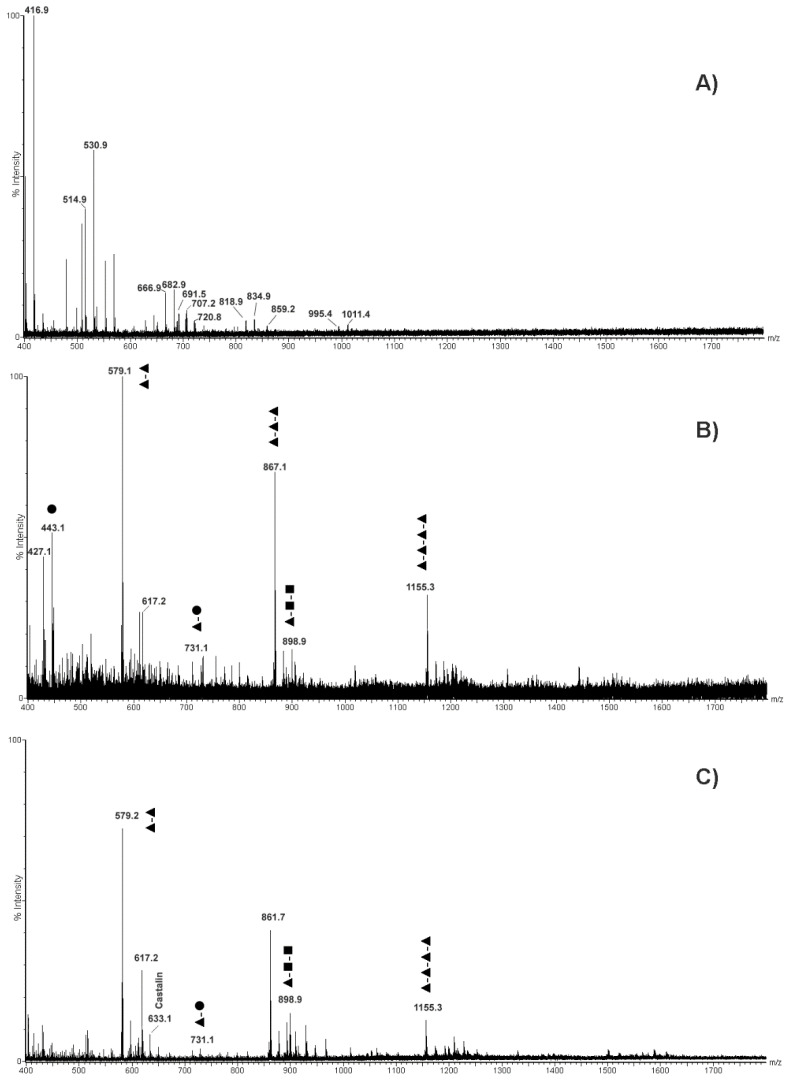
MS spectra of the polyphenol compounds recorded in positive ion mode occurring in the wastewater samples recovered at the end of three curing assays: (**A**) water containing cultured *Lb. pentosus*; (**B**) tap water; (**C**) recycled wastewater.

## Data Availability

The data presented in this study are available in Appendix A.

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
