# Peer review of "Polyphenol Profiling of Chestnut Pericarp, Integument and Curing Water Extracts to Qualify These Food By-Products as a Source of Antioxidants"

_molecules, 2021, doi:10.3390/molecules26082335_

Round 1

Reviewer 1 Report

Dear Authors,

After the review process, I have several comments: you should clearly present the aim of the paper in the abstract and at the end of the introduction section; you should include numerical data in the abstract; the authors should include which application? - abstract, the last line; you should include new data in the abstract, part of them are too old, more than five years; you should present data about bioavailability and bioactivities of polyphenols after in vitro digestion, as limiting factor after consumption; Also, you should comment on the bioactive potential of functional products and bioavailability of phenolic compounds, new studies was published and have relevance for this paper; you should include a statistical section at the end of Materials and Methods; you should include and comments the results compared with a control, e.g. Figure 5; you should present limitations of the study.

Best regards!

Author Response

Reviewer: After the review process, I have several comments: you should clearly present the aim of the paper in the abstract and at the end of the introduction section;

Answer: The abstract and the introduction section present in the original version already contained the aim of the paper. To address referee’s comments, these sections have been revised to maximize this issue.

Reviewer: you should include numerical data in the abstract.

The abstract section has been modified to include numerical data, as requested.

Reviewer: the authors should include which application?

Answer: The abstract and the introduction section present in the original version already contained the possible application. To address referee’s comments, these sections have been revised to maximize this issue.

Reviewer: abstract, the last line; you should include new data in the abstract, part of them are too old, more than five years; you should present data about bioavailability and bioactivities of polyphenols after in vitro digestion, as limiting factor after consumption;

Answer: The whole manuscript has been revised according to this comment; accordingly, the Introduction section has been fully modified by adding a novel paragraph.

Reviewer: Also, you should comment on the bioactive potential of functional products and bioavailability of phenolic compounds, new studies was published and have relevance for this paper;

Answer: The whole manuscript has been revised according to this comment; accordingly, the Introduction section has been fully modified by adding findings from recent studies.

Reviewer: you should include a statistical section at the end of Materials and Methods;

Answer: A novel section on statistical evaluation of data has been added, as requested.

Reviewer: you should include and comments the results compared with a control, e.g. Figure 5; you should present limitations of the study.

Answer: Authors thank the Reviewer for the useful comment. Original Figure 5 already contained data that were subtracted of those from control samples. To address Reviewer comment, this point has been clarified in the corresponding figure legend.

Reviewer: After the review process, I have several comments: you should clearly present the aim of the paper in the abstract and at the end of the introduction section;

Answer: The abstract and the introduction section present in the original version already contained the aim of the paper. To address referee’s comments, these sections have been revised to maximize this issue.

Reviewer: you should include numerical data in the abstract.

The abstract section has been modified to include numerical data, as requested.

Reviewer: the authors should include which application?

Answer: The abstract and the introduction section present in the original version already contained the possible application. To address referee’s comments, these sections have been revised to maximize this issue.

Reviewer: abstract, the last line; you should include new data in the abstract, part of them are too old, more than five years; you should present data about bioavailability and bioactivities of polyphenols after in vitro digestion, as limiting factor after consumption;

Answer: The whole manuscript has been revised according to this comment; accordingly, the Introduction section has been fully modified by adding a novel paragraph.

Reviewer: Also, you should comment on the bioactive potential of functional products and bioavailability of phenolic compounds, new studies was published and have relevance for this paper;

Answer: The whole manuscript has been revised according to this comment; accordingly, the Introduction section has been fully modified by adding findings from recent studies.

Reviewer: you should include a statistical section at the end of Materials and Methods;

Answer: A novel section on statistical evaluation of data has been added, as requested.

Reviewer: you should include and comments the results compared with a control, e.g. Figure 5; you should present limitations of the study.

Answer: Authors thank the Reviewer for the useful comment. Original Figure 5 already contained data that were subtracted of those from control samples. To address Reviewer comment, this point has been clarified in the corresponding figure legend.

Reviewer 2 Report

Reviewer's comment on Manuscript Number: molecules-1175216

The manuscript is devoted to polyphenol profiling of chestnut pericarp, integument and curing water extracts. The subject of the manuscript falls within the scope of Molecules.

The subject of this manuscript is really interesting and should be further explored.

However, after reading this manuscript I have a few questions:

  1. Authors mentioned that “Different solvents were tested to evaluate the molecular  recovery as estimated by the overall signal-to-noise ratios measured during dedicated MS  analysis.” Why not using Folin−Ciocalteu assay which is fast, easy and cheap? This method was used to quantify total phenolic compounds, thus, could be used as an estimate for extraction efficiency?
  2. Was Folin−Ciocalteu assay validated? Data of such validation should be given.
  3. Was tap water used for curing analysed? Was it always the same?
  1. According to authors: “ In this case, time-course analysis of polyphenols migration into the water curing medium demonstrated that the highest and sudden molecular release occurred at the end of the curing process, namely at 96 h, when more than 90% of the total migration phenomenon 275 was observed” – how this phenomenon can be explained?

The work itself is quite interesting but some issues need clarification.

I recommend the paper to be published after major amendments.

Author Response

Reviewer: The manuscript is devoted to polyphenol profiling of chestnut pericarp, integument and curing water extracts. The subject of the manuscript falls within the scope of Molecules.

The subject of this manuscript is really interesting and should be further explored.

Answer: Authors thank the Reviewer for the positive comments.

However, after reading this manuscript I have a few questions:

Reviewer: Authors mentioned that “Different solvents were tested to evaluate the molecular recovery as estimated by the overall signal-to-noise ratios measured during dedicated MS  analysis.”Why not using Folin−Ciocalteu assay which is fast, easy and cheap? This method was used to quantify total phenolic compounds, thus, could be used as an estimate for extraction efficiency?

Answer: The whole experimental section describing polyphenol extraction has been amended to address Reviewer comments. This novel manuscript version now clarify that the Folin−Ciocalteu assay was used to this purpose, together with the response during mass spectrometric analysis.

Reviewer: Was Folin−Ciocalteu assay validated? Data of such validation should be given.

Answer: Information on the validation of the quantitative measurements (blank samples, standard curve, blind samples, etc.) has now been reported in the amended manuscript.

Reviewer: Was tap water used for curing analysed? Was it always the same?

The whole experimental section describing polyphenol extraction has been amended to address Reviewer comments. The tap water used for different curing processes was always the same; the curing processes were performed in parallel; samples of original tap water were stored and used as blank samples.

Reviewer: According to authors: “ In this case, time-course analysis of polyphenols migration into the water curing medium demonstrated that the highest and sudden molecular release occurred at the end of the curing process, namely at 96 h, when more than 90% of the total migration phenomenon was observed” – how this phenomenon can be explained?

Answer: The observed phenomenon can be tentatively justified based on what reported by Migliorini et al. [33]. These authors described that the efficiency of the water curing method depends on partial lactic/alcoholic fermentations as well as on the soaking process, which take place in parallel during the whole treatment. In particular, water soaking can per se affect the permeability of the chestnut pericarp and, as a result, may increase the solubility of polyphenols present in the inner fruit. To yield a significant polyphenol extraction, water needs to: i) permeate in the inner parts of the fruit; ii) extract metabolites; and iii) move them toward the external medium. This process requests optimal times for the different steps described above, whose individual contribution and time-course (affected by various physico-chemical parameters) is unknown at present.

The work itself is quite interesting but some issues need clarification. I recommend the paper to be published after major amendments.

Answer: Authors believe that the amended manuscript addressed all requested clarifications.

Reviewer 3 Report

In the present work, metabolites from chestnuts were extracted from pericarp and integument tissues or released in the medium from the water curing process were analyzed by MALDI-TOF-MS and ESI-qTOF-MS. As result, different polyphenols were identified and their chemical profile demonstrated that bacterial presence in water hampered the release of pericarp metabolites. Also, this study provides a rationale for traditional processing practices on fruit appearance and qualifies the corresponding wastes as a source of bioactive compounds for other applications. Based on the obtained results, I suggest that this review must be accepted to publication in Molecules after minor revision, as follow:
1.    Lines 42/43 - gallic and ellagic acids are not flavonoids. Please, revise!
2.    Line 98 – the correct is qTOF and not Q-TOF – revise this point in the manuscript.
3.    Lines 111/112 – “The mixture acetonitrile/methanol/water 2:2:1 v/v/v was identified as the optimal one for extraction of polyphenols”. Why the authors decided to use this mixture of solvents to extraction of polyphenols? A reference must be included here.
4.    It’s not clear to this referee how the pericarps were obtained. Are they used in fresh form or were dried? Detailed information must be included in the revised version of this manuscript. 
5.    Identification of compounds was performed based on the fragmentation of MS spectra. In the case of the most important compounds, a figure indicating these fragments should be presented in the manuscript together with the MS spectra.
6.    Authors report the use of ESI-qTOF-MS in high resolution – to the best of my knowledge, it means that an exact molecular formula (for a molecule or fragment) could be established. However, figure 3 showed the spectra on low resolution. Could the authors explain this difference? 
7.    Figure 4 must be reprocessed - it is so confused with much information. Please, include exclusively the most important data and results. 
8.    Line 242 and in several other parts of the manuscript – m/z must be presented in italic

Author Response

Reviewer: In the present work, metabolites from chestnuts were extracted from pericarp and integument tissues or released in the medium from the water curing process were analyzed by MALDI-TOF-MS and ESI-qTOF-MS. As result, different polyphenols were identified and their chemical profile demonstrated that bacterial presence in water hampered the release of pericarp metabolites. Also, this study provides a rationale for traditional processing practices on fruit appearance and qualifies the corresponding wastes as a source of bioactive compounds for other applications. Based on the obtained results, I suggest that this review must be accepted to publication in Molecules after minor revision

Answer: Authors thank the Reviewer for the positive comments.

Reviewer: Lines 42/43 - gallic and ellagic acids are not flavonoids. Please, revise!

Answer: Original text was amended to address this comment.
Reviewer: Line 98 – the correct is qTOF and not Q-TOF – revise this point in the manuscript.
Answer: Original text was amended to address this comment.
Reviewer: Lines 111/112 – “The mixture acetonitrile/methanol/water 2:2:1 v/v/v was identified as the optimal one for extraction of polyphenols”. Why the authors decided to use this mixture of solvents to extraction of polyphenols? A reference must be included here.

Answer: Original text was amended to address this comment.

Reviewer: It’s not clear to this referee how the pericarps were obtained. Are they used in fresh form or were dried? Detailed information must be included in the revised version of this manuscript.

Answer: Original text was amended to address this comment.
Reviewer: Identification of compounds was performed based on the fragmentation of MS spectra. In the case of the most important compounds, a figure indicating these fragments should be presented in the manuscript together with the MS spectra.

Answer: All these information were included in the original manuscript. Perhaps they escaped Reviewer attention.

Reviewer: Authors report the use of ESI-qTOF-MS in high resolution – to the best of my knowledge, it means that an exact molecular formula (for a molecule or fragment) could be established. However, figure 3 showed the spectra on low resolution. Could the authors explain this difference?

Answer: All ESI-qTOF-MS measurements were performed at a high resolution. The need of repeating experiments over the whole time of the study as well as the subtle differences observed between measured mass values overtime prompted us to report only the first decimal point of them. The complexity of the mass spectra to be annotated also prompted us to maintain this choice. Nevertheless, the availability of fragmentation spectra for most of the recorded signals as well as the recognition in the spectra of oligomeric species differing for a certain Δm value allowed us a confident molecular assignment. Worth mentioning is also that recorded mass values and consequent assignments were compared with literature data.
Reviewer: Figure 4 must be reprocessed - it is so confused with much information. Please, include exclusively the most important data and results.

Answer: Figure 4 has been maximized to highlight important data and results. Its representation in portrait format did not allowed a proper visualization. Accordingly, the figure has been changed in a landscape format.
Reviewer: Line 242 and in several other parts of the manuscript – m/z must be presented in italic.

Answer: Original text was amended to address this comment.

Round 2

Reviewer 1 Report

Dear Authors,

I do not have other comments.

Best regards!

Reviewer 2 Report

The authors have improved paper significantly, therefore, I propose the paper to be published.